# Neoadjuvant Systemic Treatment of Primary Angiosarcoma

**DOI:** 10.3390/cancers12082251

**Published:** 2020-08-12

**Authors:** Kimberley M. Heinhuis, Nikki S. IJzerman, Winette T. A. van der Graaf, Jan Martijn Kerst, Yvonne Schrage, Jos H. Beijnen, Neeltje Steeghs, Winan J. van Houdt

**Affiliations:** 1Department of Medical Oncology, Netherlands Cancer Institute, Plesmanlaan 121, 1066 CX Amsterdam, The Netherlands; n.ijzerman@nki.nl (N.S.I.); w.vd.graaf@nki.nl (W.T.A.v.d.G.); j.kerst@nki.nl (J.M.K.); n.steeghs@nki.nl (N.S.); 2Department of Medical Oncology, Erasmus MC Cancer Institute, Erasmus University Medical Center, Doctor Molewaterplein 40, 3015 GD Rotterdam, The Netherlands; 3Department of Medical Oncology, Radboudumc, Geert Grooteplein 8, 6525 GA Nijmegen, The Netherlands; 4Department of Surgical Oncology, Netherlands Cancer Institute, Plesmanlaan 121, 1066 CX Amsterdam, The Netherlands; Y.Schrage@nki.nl (Y.S.); w.v.houdt@nki.nl (W.J.v.H.); 5Department of Pharmacy & Pharmacology, The Netherlands Cancer Institute, Plesmanlaan 121, 1066 CX Amsterdam, The Netherlands; j.beijnen@nki.nl; 6Department of Pharmaceutical Sciences, Utrecht University, Universiteitsweg 99, 3584 CGUtrecht, The Netherlands

**Keywords:** sarcomas, angiosarcoma, neoadjuvant treatment, neoadjuvant therapy, outcome

## Abstract

Angiosarcoma is an extremely rare and aggressive malignancy. Standard of care of localized tumors includes surgery ± radiation. Despite this multimodal treatment, >50% of the angiosarcoma patients develop local or distant recurrent disease. The role of neoadjuvant systemic therapy is still controversial and we therefore performed a systematic review of the literature to define the role of neoadjuvant systemic therapy based on available evidence. We focused on the effects of neoadjuvant systemic therapy on: 1. The success of surgical resection and 2. the long-term survival. All articles published before October 2019 on Ovid Medline, Ovid Embase, Cochrane library and Scopus were evaluated. Eighteen case reports and six retrospective cohort studies were included. There were no randomized controlled trials. This literature showed a beneficial role of neoadjuvant chemotherapy on downsizing of the tumor resulting in an improvement of the resection margins, especially in patients with cardiac or cutaneous angiosarcoma. However, no definitive conclusions on survival can be drawn based on the available literature lacking any prospective randomized studies in this setting. We advise that neoadjuvant chemotherapy should be considered, since this could lead to less mutilating resections and a higher rate of free resection margins. An international angiosarcoma registry could help to develop guidelines for this rare disease.

## 1. Introduction

Angiosarcoma is an aggressive sarcoma subtype, mostly deriving from endothelial cells of vascular or lymphatic origin. This neoplasm most frequently arises in (sub)cutaneous blood vessels, but can arise throughout the whole body [1]. Angiosarcoma is extremely rare and accounts for less than 1% of all soft tissue sarcomas in adults with an incidence of 1.5 per 1,000,000 persons per year [2,3]. Some case reports suggest that several familial syndromes could possibly predispose for angiosarcoma, such as a mutation in the *BRCA1* or *BRCA2* gene [4,5]. 

Angiosarcomas can be divided into different subgroups, primary (sporadic) or secondary, based on the etiology of the disease [6,7]. Primary or sporadic angiosarcoma arise from progenitor or mesenchymal stem cells anywhere in the body, but seem to have a slight predilection for the breast [7,8], while secondary angiosarcomas are mostly seen on the skin because they are caused by external damage by radiation, UV-exposure or chronic lymphedema [7]. The most common variant is the UV-induced angiosarcoma, usually arising in the skin of the face and scalp (35–62%) of mainly elderly patients [1,2,9]. Radiation associated angiosarcoma can occur anywhere in the body after previous radiation but is most frequently seen in the breast after previous radiotherapy for a primary breast malignancy. It is estimated that around 1 in 10,000 patients per year previously treated for a malignancy with radiation, sooner or later develops angiosarcoma in the inflicted area [1,10]. Angiosarcoma in the extremity can be caused by chronic lymphedema and this disease is also known as Stewart-Treves syndrome [1,10]. The incidence of Stewart-Treves syndrome is between 1/10 and 1/20 of patients with cutaneous angiosarcoma [1]. Finally, several exogenous toxins are associated with the development of angiosarcoma, especially within the liver [11,12]. The separation in primary and secondary angiosarcoma is important, because there is a difference in prognosis. Patients with secondary angiosarcoma show a better median overall survival than patients with primary angiosarcoma, 20.6 vs. 7.2 months, respectively [7].

The standard of care for resectable localized disease is complete surgical resection. Despite this treatment, more than 50% of patients develop local (26–54%) or distant (>50%) recurrent disease [13,14] and only 60% of patients who initially present with localized disease survive for more than 5 years [15], meaning there is an urgent need to improve the treatment. Given this high-risk and poor prognosis of angiosarcoma, ESMO guidelines state that neoadjuvant radiation and chemotherapy may be considered [16]. Current practice regarding (neo)adjuvant treatment, however, varies widely per country and per institution. Then again, conclusive data regarding the response rates and potential survival benefit of (neo)adjuvant chemotherapy is lacking, and in modern times neoadjuvant chemotherapy is often preferred over adjuvant chemotherapy to enable response evaluation and change chemotherapy regimen when no response is observed.

In general, goals of neoadjuvant systemic treatment are: (1) to facilitate adequate surgical resection by downsizing the tumor and (2) to improve survival by treating distant micrometastases, preventing outgrowth of these metastases into macrometastases. The addition of neoadjuvant systemic therapy to angiosarcoma treatment, however, is based on relatively limited available data, and consists mostly of retrospective studies and case reports. Designing a large randomized study analyzing neoadjuvant systemic therapy for angiosarcoma would be challenging, given the rarity of the disease and the different angiosarcoma subtypes with different biological behavior. With this review, we aim to provide a summary of the current literature on neoadjuvant systemic treatment of angiosarcoma. Furthermore, we aim to analyze outcome and response rates of neoadjuvant systemic therapy and evaluate tumor resectability after neoadjuvant systemic therapy. Recommendations based on available literature are given.

## 2. Results

The literature search resulted in six retrospective cohort studies and eighteen case reports with 21 individual cases discussing neoadjuvant systemic treatment (Figure 1). Table 1 and Table 2 give an overview of the short-term and long-term outcome and of the effect of neoadjuvant systemic treatment on surgical margins of angiosarcoma patients in these studies. The retrospective cohort studies will first be discussed in more detail. The six retrospective cohort studies consist of one study with angiosarcoma of the face and scalp only, two studies discussing all cutaneous angiosarcoma, two studies discussing cardiac angiosarcoma and one study discussing all kinds of angiosarcoma. Secondly, the case reports will be discussed per tumor localization, because the site of origin of the disease affects the prognosis [14,15]. 

### 2.1. Retrospective Cohort Studies

#### 2.1.1. UV-Induced Angiosarcoma of the Face and Scalp

One of the cohort studies focused on patients with UV-induced angiosarcoma of the face and scalp. In the cohort published by Guadagnolo et al., 70 patients with angiosarcoma of the face and scalp were included of whom 44 patients (63%) received chemotherapy (33 neoadjuvant and 11 adjuvant). The addition of chemotherapy to the standard treatment was independent of the size of the tumor and most patients received the combination of gemcitabine and docetaxel or paclitaxel single agent. From the 33 patients with neoadjuvant chemotherapy, eleven patients showed a clinical CR (33%), eighteen patients a PR (55%), two patients a SD (6%) and two patients PD (6%). Nine of the patients treated with neoadjuvant chemotherapy also received adjuvant chemotherapy (27%) [17]. In this study neither the status of the resection margins, nor the addition of chemotherapy had an influence on the OS or DSS when compared to the patients who did not receive chemotherapy. Neoadjuvant chemotherapy slightly improved the 5-year distant metastases free survival (38% (*n* = 33) vs. 69% (*n* = 37), *p* = 0.06), but did not improve the local control after surgery [17].

In summary, based on this limited sample size with an unknown patient selection for neoadjuvant chemotherapy, no conclusions can be drawn on the effect of neoadjuvant chemotherapy on the local and distant control rate of UV-induced angiosarcoma of the face and scalp. However, response rates were relatively high with only 6% PD during chemotherapy.

#### 2.1.2. Cutaneous Angiosarcoma

While the current European guideline [16] does not provide strict guidance for the use of neoadjuvant chemotherapy, it was already implemented as standard of care for cutaneous angiosarcoma in the Roswell Park Center since 2008 [13]. Neoadjuvant chemotherapy is used to treat occult micrometastases and to identify patients who would not benefit from a potentially morbid surgery. Patients who develop metastases or with rapid PD during neoadjuvant chemotherapy, would be excluded from extensive surgery. Oxenberg et al. retrospectively compared data from patients with neoadjuvant chemotherapy and surgery with surgery alone [13]. They included 25 patients treated between 1996–2012 with cutaneous angiosarcoma at different locations, including breast and head and neck. From these patients, thirteen patients had a primary resection and twelve patients were treated more recently and started with neoadjuvant chemotherapy of whom eventually ten patients underwent surgery. Two patients, who developed distant metastases during neoadjuvant chemotherapy, were excluded from further comparisons. The response and outcome analyses were performed for the two subgroups as total (surgery alone (*n* = 13) vs. neoadjuvant chemotherapy and surgery (*n* = 10)), despite the heterogeneity of tumor localizations and the difference in follow-up time within the groups. Two different chemotherapeutic regimens were given: paclitaxel (*n* = 6) or gemcitabine plus docetaxel (*n* = 4). There were no differences in resection margins or type of wound closure between the two groups. Thirty percent of the neoadjuvant chemotherapy cohort had a pathologic CR (pCR), however, neoadjuvant chemotherapy did not improve the local RFS, distant DSS, DSS or OS [13]. On the other hand, delay in surgery due to neoadjuvant chemotherapy did not negatively influence the outcome of these patients either.

Sinnamon et al. searched a large national database and included 821 patients with localized cutaneous and soft tissue angiosarcoma, who underwent surgery [18]. They excluded patients who died within 90 days after surgery, which could have confounded the results. Of the 821 patients, 26% was located in the head and neck region. Overall, only 38 patients (5%) received neoadjuvant chemotherapy, but the rationale for choosing neoadjuvant treatment in these patients was not specified. Nevertheless, both neoadjuvant (median OS 3.1 years, *n* = 38) and adjuvant chemotherapy (median OS 3.8 years, *n* = 128) did not improve the median OS compared to the median OS of patients without chemotherapy (3.4 years, *n* = 655) [18]. Of note, the results could be biased, because patients with a worse prognosis, caused by larger tumors or tumors which are located in areas which are difficult to operate, are more likely to receive neoadjuvant treatment. Furthermore, no information about the chosen regimen was provided, which makes it complicated to interpret these results, because the type of chemotherapy could also affect the outcome of patients. The large number of patients in this cohort created the opportunity to identify factors associated with poor OS using Cox proportional hazards modeling. Factors significantly associated with poor survival, with descending hazard ratio (HR), were tumor size > 7 cm (HR 2.37), age > 70 years (HR 2.02), Afro-American race (HR 1.92), tumor size 3–7 cm (HR 1.64), positive resection margins (microscopic HR 1.59, macroscopic HR 3.38), grade 3 tumor (HR 1.52) and head and neck as primary localization (HR 1.44) [18].

To conclude, both cohort studies investigated the effect of neoadjuvant chemotherapy in patients with non-metastatic cutaneous or soft tissue angiosarcoma and found no survival benefit, but also no dismal effects of delaying the resection.

#### 2.1.3. Cardiac Angiosarcomas

Two of the retrospective cohort studies investigated cardiac sarcomas. Li et al. focused on the survival after a heart transplantation as an uncommon treatment of unresectable non-metastatic cardiac sarcomas in six cases from their own institute and 40 patients from the literature [19]. Among the 46 patients receiving heart transplantation for primary cardiac sarcoma, angiosarcoma was the most common histologic subtype (*n* = 14, 30%). The 46 patients with a heart transplantation were compared to seven patients with unresectable, non-metastatic cardiac sarcomas of the same institute who only received palliative treatment (systemic therapy or radiotherapy), due to patient choice or unavailability of a donor heart [19]. They found that the survival after heart transplantation was worse for angiosarcomas than other cardiac sarcomas (9 vs. 36 months, *p* = 0.002) and the survival after heart transplantation was comparable to patients receiving palliative systemic treatment only (9 vs. 8 months, *p* = 0.912) [19]. Furthermore, neoadjuvant as well as adjuvant chemotherapy did not improve the survival for all cardiac sarcoma patients (15 vs. 18 months, *p* = 0.210, and 15 vs. 26 months, *p* = 0.088, respectively) [19]. However, the rationale for the addition of neoadjuvant chemotherapy was not given in the manuscript.

Abu Saleh et al. have previously shown that in the treatment of cardiac sarcomas R0 resection margins resulted in better OS, but this was not easily achieved [20]. They hypothesize that neoadjuvant chemotherapy could result in debulking of the tumor and therefore could aid in achieving negative margins during surgery. They included 44 cardiac sarcoma patients of whom the majority had angiosarcoma (*n* = 30, 68%). As part of a clinical trial to investigate the effect of neoadjuvant chemotherapy on the survival, 32 patients received neoadjuvant chemotherapy, of which 24 (80%) patients with angiosarcoma. The demographic characteristics were comparable between the group who received neoadjuvant chemotherapy and the group who received no chemotherapy. However, stage at start of treatment differed between the groups, 63% of the patients in the neoadjuvant group had distant metastases and only 33% in the group treated without chemotherapy (*p* = 0.082). The first line neoadjuvant treatment of the sarcoma patients consisted of doxorubicin plus ifosfamide and the second line consisted of gemcitabine plus docetaxel. Both patients with local and limited metastasized disease were included, if they were considered eligible for surgery. An R0 resection resulted in a five times longer median survival and neoadjuvant chemotherapy doubled the R0 resection rate (24% vs. 61%, *p* = 0.03) [20]. The 30-day mortality rate was lower in the group who received neoadjuvant chemotherapy but not significantly (3 vs. 8%, *p* = 0.476) and there was no difference in 30-day postoperative complications [20].

Based on these two retrospective cohort studies, the addition of neoadjuvant chemotherapy to resection of the tumor could be a preferable therapeutic approach with a good safety profile and an improved R0 resection rate in a selective patient group of operable cardiac angiosarcoma. In inoperable non-metastatic cardiac sarcoma patients, a heart transplantation with or without neoadjuvant or adjuvant chemotherapy does not result in a survival benefit.

#### 2.1.4. Other

The group of Fayette et al. looked into a dataset of 164 patients with all the different histological angiosarcoma subtypes [14]. From these patients, data regarding systemic treatment was available of 144 patients. Seventeen patients received chemotherapy after R2 resection or for inoperable disease, with a 59% response rate (18% CR, 41% PR), 12% SD and 29% PD during treatment. The demographic characteristics of the different treatment groups were not compared in this study. Treatment regimens were either doxorubicin alone, ifosfamide alone or a combination of doxorubicin with ifosfamide. However, chemotherapy did not result in a significant difference in OS or PFS [14]. Smaller tumor size (<5 cm), histological grade (low and no necrosis) and R0 resections were associated with a better OS [14]. Neither the rationale for the addition of neoadjuvant chemotherapy to standard treatment, nor the precise response rate of the patients who received neoadjuvant chemotherapy was provided [14]. Furthermore, another chemotherapy regimen could have resulted in more activity as most current studies use a taxane based regimen.

#### 2.1.5. Conclusions Retrospective Cohort Studies

The cohort studies consisted of very heterogeneous patient groups, treated with various regimens. Patient selection for neoadjuvant chemotherapy was often not substantiated, allowing potential selection bias. Therefore, no definite conclusions regarding outcome benefit for these patients can be drawn based upon this data.

However, within these cohort studies, with heterogeneous treatment regimens and follow-up periods, the response rate (PR or CR) to neoadjuvant chemotherapy was extremely high with 88% for face and scalp angiosarcoma [13,14,17,18,19,20].

### 2.2. Case Reports

In total, eighteen case reports describing 21 patients were previously published in the literature. In this review, we will discuss the cases per tumor localization (Table 2). Potential publication bias should be considered.

#### 2.2.1. Angiosarcoma of the Breast

Eight cases of angiosarcoma of the breast have been reported of which three especially describe radiation induced angiosarcoma [23,24,25,26,27,28]. The patients were treated with a variety of chemotherapy schedules and all patients showed a response (CR or PR) to neoadjuvant chemotherapy.

From the five patients with primary angiosarcoma of the breast [18,19,20,26,27], one patient had a PR (50% tumor reduction after ifosfamide/vincristine/dactinomycin) [21] and three patients had a pCR: two after treatment with gemcitabine and docetaxel and one after cisplatin/doxorubicin/paclitaxel given concurrently with thalidomide [18,19,27]. No short term follow-up data on tumor reduction was available for the fifth case, but the patient was disease free 15 months after neoadjuvant therapy with an injection of cyclophosphamide/5-FU into the artery that supplied the tumor and surgery [22]. None of the patients had recurrent disease during the reported follow-up period (range 0.5–2 years).

Of the patients with radiation induced angiosarcoma [26,27,28], all patients were treated with neoadjuvant gemcitabine, two combined with docetaxel [27,28] and one combined with carboplatin [26]. Each patient showed clinical improvement after chemotherapy and the two patients with available follow-up data were disease free after 9 months and 1 year, respectively [27,28].

In these cases, angiosarcoma of the breast was quite sensitive to chemotherapy with clinical responses in all patients. All patients were disease free after a follow-up of 6–24 months. Of note, non-responding patients are generally not overrepresented in case reports.

#### 2.2.2. Angiosarcoma of the Face and Scalp

Three case reports (four patients) elaborate on neoadjuvant systemic therapy in UV-induced cutaneous angiosarcoma of the face and scalp [29,30,31]. Two patients had a pCR after treatment with bevacizumab and radiotherapy. After 8 and 26 months of follow-up the patients were still disease free [29]. One patient did not show a response after five cycles of paclitaxel but had a remarkable response on photodynamic therapy and was still recurrence free after 6 months [30]. The third case report describes a patient who had a decline of the tumor size after treatment with cisplatin plus docetaxel plus 5-FU, but unfortunately developed distant metastases shortly after the surgery [31].

In summary, three out of four patients with UV-induced angiosarcoma of the face and scalp had a response to neoadjuvant chemotherapy. Three patients were disease free after 6–26 months, one had metastatic disease shortly after surgery. Finally, the study from des Guetz et al. [32] describes three patients with radiotherapy associated angiosarcoma who were treated with neoadjuvant chemotherapy. One of these patients had a PR after neoadjuvant chemotherapy. Which chemotherapy regimen this patient received, was not specified.

#### 2.2.3. Cardiac Angiosarcoma

Two case reports describe patients with cardiac angiosarcoma [33,34]. All patients received doxorubicin based regimens to enhance the resectability of the tumors followed by resection of the tumor in one patient [34] and a heart transplantation in another patient [33]. In one patient neoadjuvant treatment was used to downstage the disease to enable surgery [34]. All patients showed a positive response to neoadjuvant chemotherapy and are disease free after a follow-up period of 24–33 months [33].

#### 2.2.4. Angiosarcoma with other Origin

The remaining case reports describe four cases with a histologically proven angiosarcoma with rare sites of origin:, one in the spleen [35], one in the calvarial space [36], one in the seminal vesicle [37], and one in the thyroid [38]. These four patients received a variety of neoadjuvant therapies, which makes it difficult to interpret the impact of these separate cases for a general treatment advise. Almost all patients showed a response to chemotherapy and all patients showed long term disease control after surgery [35,36,37,38].

## 3. Discussion

Given the often dismal prognosis of angiosarcoma, neoadjuvant systemic therapy is increasingly being considered as a valid treatment option to downsize the tumor, facilitating adequate surgical resection, but also to evaluate tumor biology to prevent unnecessary extensive surgery in case of early metastases, and prolong survival. However, literature discussing neoadjuvant strategies is limited, as we show in this systematic review. The cohort studies (Table 1) consisted of heterogeneous patient groups with low patient numbers and included both -prognostically different- primary and secondary angiosarcomas [7], patients treated with various treatment regimens and with different follow-up periods. Neoadjuvant chemotherapy was more often added to the standard treatment of recently diagnosed patients. Patient selection for neoadjuvant chemotherapy was often not substantiated and therefore, there will almost certainly have been a selection bias. And lastly, with the improvement of current histological diagnostics, some of the more previously diagnosed angiosarcomas are probably not real angiosarcomas, but other vascular tumors [7]. The study of Weidema et al. even showed that 16% of the angiosarcoma patients was wrongly classified as angiosarcoma after reevaluation of the histological material, however with a clear improvement since the introduction of molecular diagnostics [7]. Therefore, no definite conclusions can be drawn based on these data. Nevertheless, within this retrospective cohort studies with heterogeneous treatment regimens, the response rate (PR or CR) after neoadjuvant chemotherapy was very high for face and scalp angiosarcoma. No survival benefits were seen after neoadjuvant chemotherapy, although, in fact, this can only be assessed properly in randomized trials, which are lacking.

Because of the retrospective nature of the studies, these results should be interpreted with caution. Patient numbers are low and a wide diversity of chemotherapeutic regimens were investigated within different tumor sites of origin. Besides, patients with locally, primary and recurrent disease were all included, despite the influence of these characteristics on the outcome of angiosarcoma patients. Theoretically, patients with recurrent disease, with lymph node dissections or with inoperable disease might benefit more from neoadjuvant chemotherapy than patients with primary angiosarcoma. Most importantly, there is an enormous selection bias in patients receiving neoadjuvant chemotherapy, since neoadjuvant chemotherapy is not standard of care in most hospitals. Patients with more advanced disease and high-risk disease are probably selected for neoadjuvant chemotherapy, which impacts the interpretation of survival comparisons with smaller, primary resectable, tumors. Selection bias could also have occurred the other way around, since mostly younger and more fit patients are selected for neoadjuvant chemotherapy, because they can manage the treatment toxicity better, which would result in an overestimation of overall survival. Unfortunately, in most studies, the rationale for patient selection was not discussed and it is therefore extremely difficult to draw any conclusions on the effect of chemotherapy on survival.

Additionally, the type of angiosarcoma influences the outcome. For instance, patients with cardiac or visceral angiosarcoma have a worse prognosis compared to patients with cutaneous angiosarcoma [7,19,20,27]. Patients with cutaneous UV-induced angiosarcoma have a relatively better survival, despite the challenge in getting clear surgical margins [18]. Furthermore, these patients often have multi-satellite disease [17]. In patients with angiosarcoma of the scalp, the aim of neoadjuvant chemotherapy could primarily be to achieve less (mutilating) surgeries rather than achieving prolonged survival. Prognostic factors which were independently correlated with a worse prognosis were positive resection margins, primary location on the face or scalp, tumor size (>5.0 cm), grade 3 histology, multi-satellite disease, older age (>70 years), primary angiosarcoma, Afro-American ethnicity, metastatic disease and worse performance status [7,13,14,15,18,20]. All these prognostic factors should be taken into account to make a clean interpretation of the effect of the addition of neoadjuvant treatment to the standard of care.

Another important conclusion, also highlighted by Oxenberg et al., is that any delay in surgery caused by neoadjuvant chemotherapy did not seem to influence the outcome, since there was no difference in outcome between patients who received neoadjuvant chemotherapy and patients who did not, despite the fact that two patients were progressive under chemotherapy and did not receive the planned surgery [13]. Additionally, neoadjuvant chemotherapy could offer additional time to observe the tumor biology and identify these progressive patients who would not benefit from aggressive surgery [13]. Furthermore, we did not find any studies reporting a worse outcome with the addition of neoadjuvant chemotherapy. Therefore, the addition of a neoadjuvant treatment to standard of care could be a safe and individualized option for a selected group of patients [13,20].

Lastly, it is unclear which chemotherapy regimen is giving the best results in angiosarcoma in general. Despite excellent short-term responses, the benefit for the long-term outcome is debatable. Taxanes, doxorubicin and gemcitabine regimens all report responses, but alternatives may be considered. For example, because of the high expression of beta-receptors on vascular tumors the addition of the β-blocker propranolol to chemotherapy-based regimens might be beneficial according to literature [39,40,41]. Furthermore, newer drugs such as checkpoint inhibitors, have shown relatively good responses in especially the UV-induced angiosarcoma, making this a potential drug to use in the neoadjuvant and metastatic setting [1,41,42,43,44]. In particular in elderly with cutaneous angiosarcoma paclitaxel may give durable responses [45]. Currently there is one recruiting study in which paclitaxel is combined with chemoradiation as induction treatment of cutaneous angiosarcoma (NCT03921008).

To summarize, there are several limitations of this review which are important for the interpretation of the results. Current literature only consisted of retrospective studies of heterogeneous patient populations with low patient numbers, treated with various regimens and lacking the rationale for treatment choice or evaluation of possible confounders in treatment response. Considering these limitations and the challenges in performing a randomized controlled trial in a rare tumor type, an international registry with data on angiosarcoma could be a very valuable source of information. An easily accessible registry could help to develop international treatment guidelines, identify new treatment targets and elucidate angiosarcoma characteristics. Recently in the US, a project was set up to collect angiosarcoma patient data. Patients are approached via social media and patient advocacy groups and give their consent online, making it a very innovative patient-partnered approach [46]. Expansion of this kind of databases to other countries, would help in the design and execution of new randomized trials, to increase patient numbers, and provide internationally accepted treatment guidelines. But the challenge of data protection is certainly something that needs to be addressed.

## 4. Materials and Methods

A search was performed in Ovid Medline, Ovid Embase, Cochrane library and Scopus with thesaurus terms and words in title, abstract and (author) keywords. We searched for angiosarcoma, hemangiosarcoma and lymphangiosarcoma in combination with terms for ‘neoadjuvant therapy’, ‘preoperative therapy’, ‘targeted therapy’ and ‘immunotherapy’. The searches were performed on 25 October 2019. We applied no limits in publication date. Additional articles were included using citation snowballing. Selection of relevant studies was performed independently by two authors. Conflicts in the selection of relevant articles were resolved by discussion. All studies that evaluated the effect of neoadjuvant systemic therapy in the treatment of primary, secondary or recurrent angiosarcoma on the resection margins and the long-term survival were eligible. A quality assessment was performed using the Newcastle Ottawa scale for cohort studies (Appendix A) [47].

In this systematic review the terms complete response (CR), partial response (PR) and progressive disease (PD) refer to the terms as defined in the Response Evaluation Criteria in Solid Tumors (RECIST) [48] and were mostly measured clinically. Outcome was given in terms of disease- free interval (DFI), progression free survival (PFS), disease specific survival (DSS) and overall survival (OS).

## 5. Conclusions

Unfortunately, no definitive conclusions can be drawn regarding the outcome benefit of neoadjuvant chemotherapy in patients with angiosarcoma based on the current literature. All available studies were retrospective with heterogeneous, small patient groups and diverse treatment regimens with the inherent limitations. Keeping these limitations in mind, however, the retrospective cohort studies and case reports suggest that angiosarcoma is relatively sensitive to chemotherapy (response rate of 88–93% in patients with angiosarcoma of face and scalp). Neoadjuvant chemotherapy could therefore probably be used to downsize the tumor. This downsizing could result in more resections with curative intend, less mutilating resections and a higher R0 resection rate (an increase of 5–14% of all angiosarcomas to even 50% of cardiac angiosarcomas). The studies show no clear survival benefit. Nevertheless, there is an urgent need for more studies addressing the role of neoadjuvant systemic therapy in angiosarcoma and an international angiosarcoma registry could help to develop guidelines.

### Recommendations Based on This Review of the Literature

Neoadjuvant chemotherapy could be considered to downsize the tumor, since this could lead to less mutilating resections and a higher R0 resection rate.There is no survival benefit, but also no evidence of detriment of neoadjuvant chemotherapy.There is currently no evidence of the best possible chemotherapy regimen and apart from age of the patient, also the subtype may help define the treatment choice. In particularly for UV-exposed scalp angiosarcoma in elderly, paclitaxel is generally well tolerated and more recently also checkpoint inhibitors are showing interesting responses.An international angiosarcoma registry should be set up to collect all available data on angiosarcoma patients and help to develop guidelines.

## Figures and Tables

**Figure 1 cancers-12-02251-f001:**
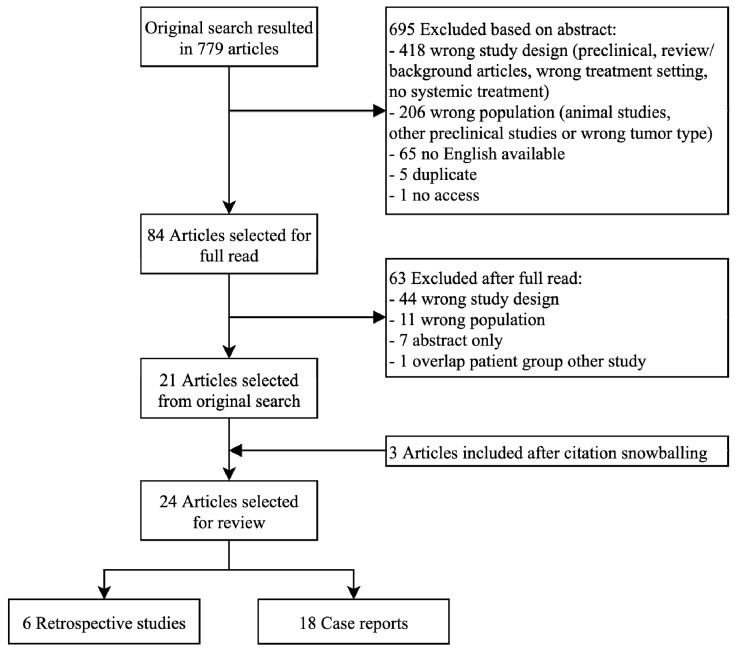
Schematic overview of search strategy.

**Table 1 cancers-12-02251-t001:** Overview of responses to neoadjuvant systemic treatment in angiosarcoma patients – retrospective cohort studies.

Refs	No. of Patients	Neoadjuvant Treatment	Patient Characteristics	Influence on Resectability	Short-Term Response	Long-Term Response
[17]	33	10 pts docetaxel + gemcitabine 5 pts paclitaxel 18 pts had diverse regimens consisting of doxorubicin + ifosfamide, cyclophosphamide + doxorubicin + dacarbazine, interferon, vincristine, doxorubicin + paclitaxel or other combinations	70 pts with non-metastatic AS of face and scalp -33 pts had NAC (regimen per pt. was ns) 20 pts had AC 9 pts had both	ns	88% response: 11 pts had CR and 18 pts had PR 2 pts had SD (6%) 2 pts had PD (6%)	Chemotherapy was not associated with a significant difference in OS or DSS, local or distant recurrence compared to pts who did not received chemotherapy
[13]	12	12 pts had ≥2 cycles of NAC: - Paclitaxel (n = 6) - Gemcitabine + docetaxel (n = 4) - not specified for 2 pts	23 pts with primary cutaneous or soft tissue AS	80% R0 resections after NAC (vs. 85% surgery alone)	30% had pCR (n = 3, one paclitaxel, two gemcitabine+ docetaxel) PR not specified -2 PD during NAC (both paclitaxel)	No statistically significant survival benefit in pts who received NAC when compared to pts who did not receive NAC
[18]	38	38 pts had NAC: site of origin AS and regimens were ns 21 pts had RT	821 localized AS	ns	No short-term FU data available	Neither RT nor chemotherapy improved the OS
[19]	10	10 pts had NAC: regimens were ns	46 pts with primary cardiac sarcomas who underwent heart transplantation-(16 pts had AS)	ns	No short-term FU data available	NAC did not provide survival benefit after heart transplantation compared to pts who only received heart transplantation
[20]	24	Median of 6 cycles of doxorubicin + ifosfamide or gemcitabine + docetaxel	32 pts with right sided heart sarcoma had NAC (24 with AS)	47% R0 resections after NAC (vs. 33% surgery alone)	No significant difference in the 30-day postoperative outcomes	Median survival 20 months with NAC vs. 9.5 months without NAC (*p* = 0.417). Median survival higher after R0 resection (53.5 vs. 9.5 months positive margins, *p* = 0.004)
[14]	17	Doxorubicin +/− ifosfamide	9 pts received NAC 7 pts after R2 resection or for inoperable disease	ns	3 pts had CR (18%) 7 pts PR (41%) 2 pts SD (12%) 5 pts PD (29%)	No significant differences in OS or PFS between pts who received NAC compared to pts without NAC

AC = adjuvant chemotherapy, AS = angiosarcoma, CR = complete response, DFI = disease-free interval, DSS = disease specific survival, FU = follow up, NAC = neoadjuvant chemotherapy, No = number, ns = not specified, OS = overall survival, pCR = pathologic complete response, PR = partial response, pt.(s) = patient(s), refs = references, vs. =versus RT = radiotherapy.

**Table 2 cancers-12-02251-t002:** Overview of responses to neoadjuvant systemic treatment in angiosarcoma patients—case reports.

Case Report Reference	Neoadjuvant Treatment	Patient Characteristics	Short-Term Response	Long-Term Response
Primary angiosarcoma of the breast				
[21]	4 cycles of ifosfamide, vincristine and dactinomycin	1 pt.	Tumor reduction of 50%	Disease free after 2 yrs. of FU
[22]	Arterial injection with cyclophosphamide and 5-FU	1 pt.	No short-term FU data available	Disease free after 15 months of FU
[23]	Gemcitabine and docetaxel	1 pt.	pCR	No evidence of recurrence 20 months after the initial diagnosis
[24]	Gemcitabine and docetaxel	1 pt.	pCR	Disease free after 2 yrs. of FU
[25]	4 cycles of cisplatin, doxorubicin and thalidomide, followed by paclitaxel, cisplatin and thalidomide	1 pt.	pCR in the breast and axillary lymph nodes	No recurrence 6 months after the initial diagnosis
Radiation induced angiosarcoma of the breast				
[26]	4 cycles of gemcitabine and docetaxel	1 pt.	Clinical improvement after 2 cycles, near CR on MRI after 4 cycles	No FU data available
[27]	3 cycles of gemcitabine and docetaxel	1 pt.	Minimal residual disease in resected tissue	Disease free after 9 months of FU
[28]	8 cycles of carboplatin and gemcitabine	1 pt.	Improvement of local condition of the breast	No recurrence 1 yr. after the surgery
[29]	3-4 cycles of bevacizumab and RT 50 Gy	2 pts with AS of the face	pCR	Disease free after 8.5 (pt. 1) and 26 months (pt. 2) of FU
[30]	5 cycles of paclitaxel Thereafter 5× PDT	1 pt. with AS of the scalp	No metastasis, no improvement of skin lesions Improvement of all skin lesions	Disease free after 6 months of FU
[31]	1 cycle of cisplatin, docetaxel and 5-FU	1 pt. with radiation induced AS of the face	Decreased tumor size from 35 × 21 mm to 19 × 13 mm on MRI	Lung metastasis after surgery. Progressive metastasis after AT
[32]	Cyclophosphamide, vincristine, doxorubicin and dacarbazine Doxorubicin, ifosfamide and dacarbazine	3 pts with post-irradiation AS (AS location not specified)	PR 1/3 pts	No FU data available
[33]	Doxorubicin, dacarbazine, ifosfamide and mesna followed by RT 2600 cGy for 1 month	1 pt.	Not specified	Disease free after 33 months of FU
[34]	3 cycles of doxorubicin and dacarbazine	1 pt.	Tumor became operable	Disease free after 2 yrs. of FU
[35]	3 cycles paclitaxel	1 pt. with AS of the spleen	PR after 3 cycles on CT	No recurrence 14 months after start of treatment
[36]	Vincristine, cyclophosphamide and actinomycin	1 pt. with calvarial AS	PD	Disease free after 3 yrs. of FU
[37]	2 cycles of ifosfamide, doxorubicin, mitomycin, cisplatin and mesna Followed by 50 Gy and 2 cycles mitomycin, doxorubicin and cisplatin	1 pt. with AS of seminal vesicle	After 2 cycles of NAC decreased tumor size from 5.6 × 5.1 to 4.3 × 4.0 cm No significant changes after RT	Disease free after 6 yrs. of FU
[38]	1 cycle of taxol, followed by 3 cycles of gemcitabine	1 pt. with epithelioid AS of the thyroid	<10% viable tumor cells left in surgical specimen	Disease free after 70 months of FU

AS = angiosarcoma, AT = adjuvant treatment, CR = complete response, CT = computed tomography, FU = follow-up, (c)Gy = (centi)gray, HIPEC = heated (hyperthermic) intraperitoneal chemotherapy, MRI = magnetic resonance imaging, NAC = neoadjuvant chemotherapy, pCR = pathologic complete response, PD = progressive disease, PDT = photodynamic therapy, PR = partial response, pt.(s) = patient(s), RFA = radio frequent ablation, RT = radiotherapy, yr.(s), year(s).

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
