# Peer review of "Neoadjuvant Systemic Treatment of Primary Angiosarcoma"

_cancers, 2020, doi:10.3390/cancers12082251_

Round 1

Reviewer 1 Report

Nicely written review on a very rare malignancy (angiosarcoma). The authors presented the data from available studies and discussed the limitations of these studies. 

Would suggest to soften the language in the last paragraph where the authors give recommendations as it's very hard to have any conclusions for the studies they presented. A limitation they mentioned.

Author Response

Reviewer 1: Nicely written review on a very rare malignancy (angiosarcoma). The authors presented the data from available studies and discussed the limitations of these studies.

Would suggest to soften the language in the last paragraph where the authors give recommendations as it's very hard to have any conclusions for the studies they presented. A limitation they mentioned.

Response:  We appreciate your comment. It is indeed hard to formulate recommendations based on the limited literature. We have softened the language in the final paragraph by deleting “the advantage of the neoadjuvant setting is a proper evaluation of its efficacy” in line 442-447 and added “but also no evidence of detriment of neoadjuvant chemotherapy” to line 445 as was also requested by reviewer 2 and 3.

Reviewer 2 Report

I commend the authors for a very well organized and well written overview of the available literature of neoadjuvant systemic treatment of primary angiosarcoma.  The manuscript is a nice addition to the sarcoma literature and reference for those who treat and care for angiosarcoma patients. 

I have no major comments.  Only recommendations is to review line 262 for minor typographical error and line 429 would consider adding to "There is no survival benefit." perhaps a statement that there is also no evidence of detriment of neoadjuvant therapy.

Author Response

Reviewer 2: I commend the authors for a very well organized and well written overview of the available literature of neoadjuvant systemic treatment of primary angiosarcoma.  The manuscript is a nice addition to the sarcoma literature and reference for those who treat and care for angiosarcoma patients.

I have no major comments.  Only recommendations is to review line 262 for minor typographical error and line 429 would consider adding to "There is no survival benefit." perhaps a statement that there is also no evidence of detriment of neoadjuvant therapy.

Response: Thank you very much for your kind words and compliments. We have deleted line 263-265 based on the comments of reviewer 3, and changed the recommendation in line 445 into “There is no survival benefit, but also no evidence of detriment of neoadjuvant chemotherapy”.

Reviewer 3 Report

The Authors present a systematic review on the use of of neoadjuvant chemotherapy (NAC) for patients with primary angiosarcoma. In my view, the manuscript is of interest to the clinical community and summarizes well valuable information; therefore I believe it would be of interest to the readership of Cancers. However, there are some flaws that need to be addressed in the methodology of the Review. The manuscript would benefit from further revision and clarification with respect to its limitations. In addition, safety profile review of available NAC in angiosarcomas would add to the value of the present study.

Major Points:

  1. Please comply to Cochrane guidelines and use PICOs to describe the study protocoll and the outcomes of interest in the methods section.
  2.  A systematic review with or without a meta-analysis is as good as the studies included in it. Please provide quality/risk of bias assessment using the Newcastle_Ottawa scale for cohort studies.
  3. To avoid publication bias conference abstracts should also be scrutinized and considered.  No language limitation should be applied.
  4. The strength of evidence  for case reports is low and does not add to the value of the present Review and the recommendations made by the Authors.
  5.  Cochrane library and clinicaltrials.gov should also be assessed; the latter for ongoing studies on sarcomas-angiosarcomas with respect to NAC in order to discuss future options, for instance immune checkpoint inhibitors not adequately discussed here.
  6.  The limitations of the present Review with respect to tumor heterogeneity, diversity of chemotherapeutics, different confounders, problems in the design of the included studies should be confined in one paragraph in the discussion section before the conclusions of the study.
  7.  Strong recommendations, i.e. "NAC should be considered to downsize the tumor…" cannot be made based on the included studies of the present review. Please consider rephrasing and not making strong statements that are not supported by high quality evidence.
  8. Please discuss the safety profile of currently available NAC for angiosarcomas.
  9. Paragraph 2.11, lines 112-116.  Only the latest or the most relevant study from the same institution should be included to avoid overlap of the cases, unless the Authors have access to individual patient data.
  10. Please correct the databases assessed in the abstract (not only pubmed)
  11.  Please provide a figure with a proposed algorith on patient selection for NAC.

Author Response

Reviewer 3: The Authors present a systematic review on the use of of neoadjuvant chemotherapy (NAC) for patients with primary angiosarcoma. In my view, the manuscript is of interest to the clinical community and summarizes well valuable information; therefore I believe it would be of interest to the readership of Cancers. However, there are some flaws that need to be addressed in the methodology of the Review. The manuscript would benefit from further revision and clarification with respect to its limitations. In addition, safety profile review of available NAC in angiosarcomas would add to the value of the present study.

Response: Thank you for very much for your thoughtful comments. Please find below our replies in italics.

Major Points:

  • Please comply to Cochrane guidelines and use PICOs to describe the study protocol and the outcomes of interest in the methods section.
    • We have added additional information about the study protocol and eligibility criteria in line 418-422: “Selection of relevant studies was performed independently by two authors. Conflicts in the selection of relevant articles were resolved by discussion. All studies that evaluated the effect of neoadjuvant systemic therapy in the treatment of primary, secondary or recurrent angiosarcoma on the resection margins and the long term survival were eligible.”.
  • A systematic review with or without a meta-analysis is as good as the studies included in it. Please provide quality/risk of bias assessment using the Newcastle_Ottawa scale for cohort studies.
    • A quality assessment of the cohort studies was performed using the Newcastle Ottawa scale for cohort studies, as suggested, and added to the manuscript (line 421/422) and as supplemental table 1.
  • To avoid publication bias conference abstracts should also be scrutinized and considered. No language limitation should be applied.
    • Initially, we also included abstracts of conferences for this review. However, the information provided in the abstracts was too limited to properly interpret the data and incorporate them in our overview.

We consider the influence of language selection and risk for language bias considerably low. Only a very small amount of articles from the original search was non-English. Furthermore, the non-English articles which were excluded were only case reports with a limited impact on the discussion and conclusions of our review.

  • The strength of evidence for case reports is low and does not add to the value of the present Review and the recommendations made by the Authors.
    • We agree with you that no firm recommendations can be made based upon case reports. The current recommendations are primarily based on the results from the case series.
  • Cochrane library and clinicaltrials.gov should also be assessed; the latter for ongoing studies on sarcomas-angiosarcomas with respect to NAC in order to discuss future options, for instance immune checkpoint inhibitors not adequately discussed here.
    • A literature search in Cochrane Library did not result in additional eligible articles. We have added Cochrane library to the description of the methods. A search in clinicaltrials.gov resulted in one recruiting study in angiosarcoma (NCT03921008). We have added a description of this study to the discussion in line 391-393. We found no registered studies with immune checkpoint inhibition as neoadjuvant treatment of angiosarcoma.
  • The limitations of the present Review with respect to tumor heterogeneity, diversity of chemotherapeutics, different confounders, problems in the design of the included studies should be confined in one paragraph in the discussion section before the conclusions of the study
  • We have added the limitations to the final paragraph before the conclusions in line 398-402, “To summarize, there are several limitations of this review which are important for the interpretation of the results. Current literature only consists of retrospective studies of heterogeneous patient populations with low patient numbers, treated with various regimens and lacking the rationale for treatment choice or evaluation of possible confounders in treatment response".
  • Strong recommendations, i.e. "NAC should be considered to downsize the tumor…" cannot be made based on the included studies of the present review. Please consider rephrasing and not making strong statements that are not supported by high quality evidence.
    • We have softened the recommendations made in the final paragraph by deleting “ the advantage of the neoadjuvant setting is a proper evaluation of its efficacy” in line 442-447 and added “but also no evidence of detriment of neoadjuvant chemotherapy” to line 445 as was also suggested by reviewer 1 and 2.
  • Please discuss the safety profile of currently available NAC for angiosarcomas.
    • Currently there are no standard neoadjuvant chemotherapeutic regimens in the treatment of angiosarcoma and, therefore, it is hard to discuss the safety profile extensively for all different regimens.

We agree that it in clinical practice it is important to weigh the safety profile before making the final decision. A comment about the reported safety in the described cases was made in line 384-386.

  • Paragraph 2.11, lines 112-116. Only the latest or the most relevant study from the same institution should be included to avoid overlap of the cases, unless the Authors have access to individual patient data.
    • We have no access to the individual patient data, so we decided to only discuss the most recent, largest patient serie from this institute (Guadagnolo et al.) in the revised version of your review.
  • Please correct the databases assessed in the abstract (not only pubmed)
    • Thank you for your attentiveness. We have added “Ovid Medline, Ovid Embase and Scopus” in the abstract.
  • Please provide a figure with a proposed algorithm on patient selection for NAC.
    • We really appreciate your suggestion to include a proposed algorithm to our review paper. We agree that this could be a very valuable, practical tool for treating physicians around the world. However, we do not think we are able to make such an algorithm based on the limitations of available evidence.

Round 2

Reviewer 3 Report

The Authors have now revised and improved their manuscript, as requested by the reviewers. Therefore, the present version of the manuscript could be considered for publication in Cancers.